# Ambient Air Pollution Exposure Association with Anaemia Prevalence and Haemoglobin Levels in Chinese Older Adults

**DOI:** 10.3390/ijerph17093209

**Published:** 2020-05-05

**Authors:** Mona Elbarbary, Trenton Honda, Geoffrey Morgan, Yuming Guo, Yanfei Guo, Paul Kowal, Joel Negin

**Affiliations:** 1Faculty of Medicine and Health, Sydney School of Public Health, The University of Sydney, Sydney, NSW 2006, Australia; geoffrey.morgan@sydney.edu.au (G.M.); joel.negin@sydney.edu.au (J.N.); 2Department of Family and Preventive Medicine, University of Utah, Salt Lake City, UT 84108, USA; trenton.honda@utah.edu; 3School of Public Health, University Centre for Rural Health, The University of Sydney, Sydney, NSW 2006, Australia; 4Department of Epidemiology and Preventive Medicine at School of Public Health and Preventive Medicine, Monash University, Clayton, VIC 3800, Australia; yuming.guo@monash.edu; 5Shanghai Municipal Center for Disease Control and Prevention, Shanghai 200336, China; guoyanfei@scdc.sh.cn; 6School of Medicine and Public Health, The University of Newcastle, Callaghan, NSW 2308, Australia; paul.kowal@newcastle.edu.au

**Keywords:** air pollution, ageing, blood

## Abstract

Background: Health effects of air pollution on anaemia have been scarcely studied worldwide. We aimed to explore the associations of long-term exposure to ambient air pollutants with anaemia prevalence and haemoglobin levels in Chinese older adults. Methods: We used two-level linear regression models and modified Poisson regression with robust error variance to examine the associations of particulate matter (PM) and nitrogen dioxide (NO_2_) on haemoglobin concentrations and the prevalence of anaemia, respectively, among 10,611 older Chinese adults enrolled in World Health Organization (WHO) Study on global AGEing and adult health (SAGE) China. The average community exposure to ambient air pollutants (PM with an aerodynamic diameter of 10 μm or less (PM_10_), 2.5 μm or less (PM_2.5_), 1 μm or less (PM_1_) and nitrogen dioxide (NO_2_)) for each participant was estimated using a satellite-based spatial statistical model. Haemoglobin levels were measured for participants from dried blood spots. The models were controlled for confounders. Results: All the studied pollutants were significantly associated with increased anaemia prevalence in single pollutant model (e.g., the prevalence ratios associated with an increase in inter quartile range in three years moving average PM_10_ (1.05; 95% CI: 1.02–1.09), PM_2.5_ (1.11; 95% CI: 1.06–1.16), PM_1_ (1.13; 95% CI: 1.06–1.20) and NO_2_ (1.42; 95% CI: 1.34–1.49), respectively. These air pollutants were also associated with lower concentrations of haemoglobin: PM_10_ (−0.53; 95% CI: −0.67, −0.38); PM_2.5_ (−0.52; 95% CI: −0.71, −0.33); PM_1_ (−0.55; 95% CI: −0.69, −0.41); NO_2_ (−1.71; 95% CI: −1.85, −1.57) respectively. Conclusions: Air pollution exposure was significantly associated with increased prevalence of anaemia and decreased haemoglobin levels in a cohort of older Chinese adults.

## 1. Introduction

Anaemia is a significant public health problem in both industrialised and non-industrialised countries, with an estimated population prevalence of nearly one in four [1,2]. Anaemia is defined as a decrease in the concentration of circulating red blood cells (RBC) or haemoglobin concentration (below 12 g/dL in women and 13 g/dL in men), contributing to impaired oxygen transport [3,4]. Anaemia can be viewed as a constellation of numerous aetiologies, all of which result in decreased red cell mass or haemoglobin levels. Because of this, anaemia types are generally grouped into three categories by their mechanisms: (1) Nutrient deficiency (iron, vitamin B_9_ and vitamin B_12_), (2) anaemia of inflammation or anaemia of chronic disease and (3) anaemia of unknown cause [5]. In 2002, the World Health Organization (WHO) estimated that iron deficiency anaemia was one of the ten most significant factors contributing to the global burden of disease [2]. It is particularly challenging for developing countries, which have a disproportionately large anaemia disease burden. Previous studies have found that South Asia and sub-Saharan Africa account for upwards of 37.5% and 23.9%, respectively, of the global anaemia burden [6]. In China, where more than 18% of the global population lives [7], the anaemia prevalence ranges from 13.4% to 34% depending on the region [2,8,9].

Several studies have demonstrated links between anaemia and increased morbidity and mortality [10,11,12] with notably higher risk seen among older people (defined as individuals of ≥65 years of age) [13]. A systematic review of 45 studies found that 12–17% of the world’s elderly population has anaemia [14]. Moreover, around 40% of hospitalised older adults and 47% of those who are in nursing homes are anaemic [14,15]. These findings are consistent with several other cohort studies from different countries, which have shown that anaemia prevalence increases with age [15,16,17,18,19,20,21], reaching nearly 50% in men aged 80 years and older [15]. The clinical implications of anaemia are broad in this age group since the disease is often impacted by, or comorbid with, various severe medical disorders, including as myelodysplastic syndromes, cancer, chronic kidney disease and gastrointestinal diseases [22]. Anaemia can also stress the cardiovascular system, resulting in compensatory increases in cardiac output and heart rate. This effect, coupled with local tissue hypoxia resulting from decreased oxygen-carrying capacity, is thought to contribute to an increased rate of overall functional decline in older people [23]. Anaemia has also been previously associated with increased risk of cardiovascular diseases [24], cognitive impairment [25,26], insomnia [27], impaired mood [28] and poor physical performance [29]. As global demographic changes have resulted in an aging population, the total disease burden attributable to anaemia in older people will likely continue to grow [30]. 

A number of risk factors for anaemia have been described. These include low socioeconomic status, poor dietary intake [31] and low body mass index (BMI) [32]. While any of these can lead to the low haemoglobin levels and/or red cell concentrations, which are pathognomonic for anaemia, many of those risk factors frequently co-occur [33]. In addition to these individual-level risk factors, it is possible that ubiquitous environmental exposures, such as air pollution, will also play a role in the development and progression of anaemia. Ambient air pollution is a known risk factor for numerous highly prevalent, chronic diseases, including cardiovascular, mental and respiratory tract disorders [34]. Moreover, the negative impact of air pollution is especially profound in older adults [35], and the developing parts of the world, where the prevalence and severity of anaemia are known to be the highest [36].

Air pollution in developing countries has led to tremendous public health impacts [37]. Driven by economic growth, industrialisation in these regions has led to substantial increases in air pollutant levels, often an order of magnitude higher than those observed in developed countries. The latter has contributed to poor health conditions and posing an increasing threat to human lives [38]. A vivid example in this regard is the situation in China. The Chinese economy has been on the rise since the late 20th century, with rapid urbanisation occurring across much of the country [39] that has resulted in severe ambient air pollution levels [40]. Cohort studies have demonstrated that long-term exposure to air pollutants throughout China is associated with increased morbidity and mortality among different groups of the Chinese population, where the most predominant health conditions are chronic non-communicable diseases, such as lung cancer, respiratory diseases, cardiovascular diseases and stroke [41,42]. As both air pollution levels and anaemia prevalence are high in China, it is possible that the increased cardiovascular morbidity and mortality associated with air pollution exposures may be partially explained by air pollution-induced anaemia [43]. 

Studies regarding the effects of air pollution on anaemia are scarce, with most of the prior research being focused on the short-term exposures or children population [26,43,44]. Only one study examined associations between air pollution and anaemia in older adults [45]. In that study, of 4121 older Americans, reported that an IQR increase in the one-year moving average PM_2.5_ (IQR, 3.9 μg/m^3^) and NO_2_ (9.6 µg/m^3^) were associated with a 33% (prevalence ratio, (PR) 1.33; 95% CI: 1.23–1.45) and 43% (PR 1.43; 95% CI: 1.25–1.63) increased prevalence of anaemia, respectively [43]. To our knowledge, no prior studies have examined whether air pollution is associated with anaemia or haemoglobin levels in an older Chinese population.

In our current study, we aim to address these gaps in the literature by examining the associations between long-term exposure to ambient particulate matter (PM) and gaseous pollutants (nitrogen dioxide (NO_2_)) with anaemia prevalence and haemoglobin levels in an older (>50 years) Chinese population. PM associations were examined for PM_10_ (coarse particles with a diameter of less than or equal 10 μm), PM_2.5_ (particulate matter with an aerodynamic diameter less than or equal to 2.5 μm) and PM_1_ (particulate matter with an aerodynamic diameter less than or equal to 1 μm).

## 2. Methods

### 2.1. Study Population

The study on global AGEing and adult health (SAGE) is a longitudinal study run by the WHO, which aims to collect and analyse extensive nationally representative cohort data on the health and well-being of adult populations and the ageing process from middle-income countries, such as China, Ghana, Mexico, India, Russia and South Africa [46].

In our analysis, we used the data from the cross-sectional baseline survey acquired via interview from adult respondents from China during the 2007 to 2010 period (SAGE China Wave 1). A probability sampling design and a five-stage cluster sampling strategy were used. First, eight provinces from 34 eligible Chinese provinces were selected. Second, one rural county and one urban district from each of these eight provinces were randomly chosen. Third, two rural and two urban townships or communities were selected in each of the counties and districts from step 2. Fourth, from each of these townships and communities, two villages or enumerations were randomly chosen. Lastly, from each of the villages, two residential blocks were selected. All selections of the included geographic regions and districts in this multi-stage approach were performed randomly in order to minimise the possible risk of bias. Approval to conduct this study was granted through the Ethics Committee of the Chinese Centre for Disease Control and Prevention.

### 2.2. Exposure Assessment

Predicted data concentrations of PM_1_, PM_2.5_, PM_10_ and NO_2_ for China were modelled using satellite remote sensing, meteorology and land use information and other data combined with ground-monitored data on the pollutants from stations throughout China. Daily ground-level measurements of PM_1_ were obtained from the 77 stations of the China Atmosphere Watch Network (CAWNET) from September 2013 to December 2014. Daily ground-level measurements of PM_2.5_, PM_10_ and NO_2_ were obtained from 1479 stations of the China National Environmental Monitoring Centre (CNEMC) from May 2014 to December 2016. 

Details of the PM_1_ predictions have been previously published [47]. Briefly, data on aerosol optical depth (AOD) was derived from two NASA Moderate Resolution Imaging Spectroradiometers (MODIS) data processing algorithms (i.e., Deep Blue and Dark Target), and then combined using an inverse variance weighting method at 0.1 deg (approximately 10 km) grid cell resolution. Ground-monitored PM_1_ data was linked with the AOD data, vegetation data, land use information, meteorological data and other spatial data using a generalized additive model to predict daily PM_1_ grid cell concentrations (2005 to 2014). A similar approach using random forest models was implemented to predict the daily PM_2.5_ grid cell concentrations (2005 to 2016) and daily PM_10_ grid cell concentrations (2005 to 2016) and the details have been published [48,49].

Likewise, details on NO_2_ (from 2013 to 2016) prediction have also been published [50]. In summary, satellite-derived tropospheric column densities of NO_2_ (molecules/cm^2^) were obtained from the OMI-NO2 level-3 data product (OMNO2d version 3) [51] at 0.25 degree (approximately 13 × 24 km^2^) resolution. Temporal convolution with Gaussian kernels, which tended to create gentle smoothing and preserve edges, was operated to filter out noise while filling data gaps because of satellite orbit, cloudy conditions, or surface reflectance. Ground-monitored NO_2_ data was linked with satellite NO_2_ data, vegetation data, land use information, meteorological data, road density and other spatial predictors using a random forest model to predict daily NO_2_ grid cell concentrations. 

Ten-fold cross-validation was performed using the monitored data to assess the predictive ability of the models for PM_1_, PM_2.5_, PM_10_ and NO_2_. The results for the adjusted coefficient of determination (R^2^) and root-mean-squared error (RMSE) are shown in Appendix A.

We defined exposure to long-term ambient air pollution as the average PM and NO_2_ concentrations of the three years (2005–2007) prior to the survey. Finally, community locations of participants were geo-coded and then used to link the estimated annual PM and NO_2_ concentrations.

### 2.3. Outcome Assessment

Using the WHO criteria, anaemia was defined as the haemoglobin concentrations below 13 g/dL in men and below 12 g/dL in women [3]. Haemoglobin was collected via dried blood spots captured and transported on filter paper to centralised laboratories. The analysis was performed using high-performance liquid chromatography. Haemoglobin from dried blood spots is tightly correlated with venous samples in population health studies [52,53].

### 2.4. Covariates

We controlled for demographic, biometric, health behaviour, indoor air pollution, socioeconomic status and comorbid disease covariates based upon their previous associations with either air pollution exposure [54,55], anaemia risk [56,57], or both [58]. The main potential confounders, such as age and sex, were included in the SAGE China Wave 1 survey. In addition to these, height and weight were also measured to calculate BMI. Two and more daily servings of fruits, as well as three or more daily serves of vegetables, were regarded as sufficient dietary intake [59]. Alcohol consumption and smoking status were defined as never or ever. The global physical activity questionnaire was used to measure the duration, frequency and intensity of general physical activity [60]. The questionnaire included information about moderate or vigorous physical activities during work, transport activities to and from places and recreation/leisure time activities. Based on the responses from participants, we classified them into three main groups based upon the intensity and time spent on each activity, corresponding to total energy requirements in metabolic equivalents (METs): low, moderate and high physical activity [60,61].

The types of fuel used at home for cooking were classified as clean (electricity and natural gas) and unclean (coal, wood, dung and agricultural residues). The information on the most common type of fuel for cooking was important for assessing indoor air pollution. The highest educational attainment was categorised as: (1) no school, (2) primary school and (3) middle or high school. The participants’ socioeconomic status was categorised based on their self-reported household annual income: low ≤¥15,000; and high >¥15,000 using the median as the cut-off point. Finally, participants’ self-reported hypertension, chronic lung disease and arthritis were used for comorbidity assessment.

### 2.5. Statistical Analysis 

We examined the association between haemoglobin and an interquartile range (IQR) increases in three-year moving averages of PM_10_, PM_2.5_, PM_1_ and NO_2_ in single pollutant models. Haemoglobin measurements for participants in the same community may be correlated with each other, violating the independence assumption of regression models. To account this, we considered a two-level linear regression model where participants were considered as the first-level unit and the township as the second-level unit. For models, in which anaemia was treated as a dichotomous outcome, logistic regression may overestimate the prevalence ratio given anaemia’s high prevalence in the study sample; therefore, we used a modified Poisson regression with robust error variance to estimate the prevalence ratio of anaemia [62]. Also, dose-response using mixed model cubic regression splines were calculated and plotted. Selection of covariates was based upon prior, established associations with air pollution and/or known biological effects on haemoglobin metabolism. In fully adjusted models, we adjusted for age, sex, BMI, tobacco use, physical activity, education level, fruit and vegetable intake, alcohol use, type of fuel used at home and median household income. We also investigated several potential effect modifiers age, sex, smoking, alcohol and region of the survey (north and south region). Because of spatial heterogeneity between the north and south areas of China in characteristics, such as, chemical components of PM, climate and some unknown confounders, which may modify the associations between ambient air pollution and anaemia. We conducted stratified analyses by dividing the survey region into north (including the other four provinces: Shandong, Hubei, Jilin and Shaanxi) and south (including the other 4 provinces: Yunnan, Zhejiang and Shanghai and Guangdong). Effect modification was assessed by including multiplicative terms between pollutant variables in single pollutant model and some potential effect modifiers into the models. Significance of effect modification was determined if the p-value for the hypothesis test of the interaction was <0.01.

We further estimated the anaemia burden attributable to ambient NO_2_ by using three indicators: attributable cases, population attributable risk and population attributable fraction. The ambient NO_2_ level (40 μg/m^3^) set by the WHO Air Quality Guidelines was used as the reference concentration [63]. The population attributable fraction (PAF) were calculated using the formula below [64,65].

$$Population\;Attributable\;Fraction=P_{b}\left(\frac{OR-1}{OR}\right)$$

Here *P_b_* is the proportion of participants with anaemia, and OR is the prevalence ratio of anaemia in the study population. For ease of interpretation, we will present the PAF as population attribution risk percent (PAR%) by multiplying PAF by 100. To measure the adjusted contribution of NO_2_ exposure above the WHO air quality guidelines on the total risk of having anaemia the ‘adjusted PAR%’ was calculated using the adjusted ORs from the final adjusted model. We estimated 95% confidence intervals for all PAR%.

A number of sensitivity analyses were performed to ensure our findings were robust to various model specifications. First, we used average pollutants concentrations for one and five years before the baseline survey. Second, we excluded the participants with comorbidities, since respiratory and cardiovascular diseases might have had an impact on the outcomes. All analyses were conducted using STATA version 15 (StataCorp, College Station, TX, USA) and p-value < 0.05 was used to determine statistical significance.

## 3. Results

Among 15,050 participants from the SAGE China Wave 1, 13,379 were 50 years of age or older. Of these, only 10,611 (79%) individuals had valid haemoglobin measurements; 645 additional participants were excluded because of missing covariate information, including sex, age, or type of fuel used at home. As a result, 9966 participants were included in our final analyses (Appendix A). Anaemia was highly prevalent in the cohort, with 28.1% meeting the WHO definition. Participants with anaemia were significantly older than non-anaemic participants (64.3 versus 62.6 years, p < 0.01) were, exposed to higher NO_2_ levels (32.3 versus 28.4 µg/m^3^, p < 0.01), had lower BMI (23.4 versus 24.3 kg/m^2^, p < 0.01) and were more likely to have lower education levels and use clean cooking fuels (p < 0.01). Non-anaemic participants were more likely to live in rural areas, have lower physical activity level and have sufficient intake of fruit and vegetables (Table 1). There was no significant difference in sex, smoking, alcohol intake and comorbidities between these two groups.

The annual average of the four air pollutants varied across the study regions, with a range of 49–108 μg/m^3^ for PM_10_, 29–69 μg/m^3^ for PM_2.5_, 35–56 μg/m^3^ for PM_1_ and 20–46 μg/m^3^ for NO_2_. The annual average of PM_10_ and PM_2.5_ concentrations in the study regions were higher than the WHO guidelines in all eight regions of the SAGE China study (Table 2). 

Table 3 shows the associations between PM and NO_2_ and anaemia prevalence ratios. For PM_10_ and PM_2.5_, univariate models showed non-significant associations. After adjusting for potential confounding factors, these associations became positive and significant, with an IQR increase in PM_10_ (31.2 µg/m^3^), PM_2.5_ IQR (26.1 µg/m^3^) associated with a 5% (PR 1.05, 95% confidence interval (CI): 1.02–1.09) and 11% (PR 1.11; 95% CI: 1.06–1.16) increased prevalence of anaemia, respectively. Associations for PM_1_ were positive and significant in unadjusted models, with an IQR (24.1 µg/m^3^) increase associated with a PR of 1.24 (95% CI: 1.17–1.31), while covariate adjustment attenuated the associations (fully adjusted PR 1.13; 95% CI: 1.06–1.20). Associations were somewhat higher for NO_2_, with an IQR (26.8 µg/m^3^) increase associated with a 46% increased prevalence (PR 1.46; 95% CI: 1.39–1.54) in the univariate model, and were only minimally changed in fully adjusted models (PR 1.42; 95% CI: 1.34–1.49). 

Table 4 shows the statistically and clinically significant associations between PM and NO_2_ and haemoglobin levels. For each PM_10_, PM_2.5_, PM_1_ and NO_2_ IQR increase corresponded to −0.53 (95% CI: −0.67, −0.38), 0.52 (95% CI: −0.67, −0.38), 0.55 (95% CI: −0.69, −0.41) and 1.71 (95% CI: −1.85, −1.57) mg/dL decrements in haemoglobin levels, respectively. The smoothing curves of the relationships between pollutants and haemoglobin suggest non-linear relationships for the overall population (Appendix). For PM_10_ and PM_2.5_, negative associations are approximately linear until 100 and 65 (respectively), after which they become positive. In contrast, for PM_1_ the strongest negative associations occur at the lowest and highest exposures, with a curvilinear association for intermediate exposures. For NO_2_, no association is observed until 40 ug/m^3^, which is the WHO air quality guideline for NO_2_, after which they become strongly and significantly negative. 

In stratified models, the association between ambient PM and NO_2_ and haemoglobin (Appendix A) suggest that smoking may modify the association between PM exposure and haemoglobin levels, but not NO_2_. In PM_1_ and PM_10_ models, we observed a higher decrement of haemoglobin per IQR increase in smokers (−0.78 ± 0.10 vs. −0.61 ± 0.08 (p = 0.003) and −0.80 ± 0.10 vs. −0.58 ± 0.07 (p = 0.006), respectively), suggesting that smoking may enhance this association. Also, alcohol consumption showed a significant effect modification of this association for PM_1_. We observed a larger magnitude of decrease in participants who reported alcohol consumption (−0.69, β 95% CI: −0.80, −0.58, p < 0.001) than for those who reported that they never consumed alcohol (−0.61, β 95% CI: −0.68, −0.54, p < 0.001). Region of survey was also found to be an effect modifier, participants living in northern province showed higher decrement of haemoglobin per IQR increase levels compared to southern residence (−0.29 ± 0.06 vs. −0.04 ± 0.20 p < 0.001) respectively. We did not observe any significant effect modification by age, sex, household income or history of chronic obstructive pulmonary disease (COPD).

Estimates of the anaemia burden attributable to ambient NO_2_ after adjusting for confounders are summarised in Table 5. The population attributable risk associated with ambient NO_2_ higher than 40 μg/m^3^ was 4.4% (95% CI: 3.6–5.2) corresponding to 131 (95% CI: 107–155) anaemic cases in the study population. Further, we estimated the PAF and observed that approximately 26% of the anaemic cases in the study could have been averted if all older adult participants lived in areas where NO_2_ levels met the WHO air quality guidelines, assuming a causal association (Table 5). 

The associations between PM and NO_2_ and anaemia did not materially change in the sensitivity analyses (Appendix A). For example, when we used one-year and five-year average PM_1_ concentrations as the exposure variable, the PR for each IQR μg/m^3^ increase were 1.12 (95% CI: 1.05–1.21) and 1.15 (95% CI: 1.07–1.24), respectively. Additionally, when we excluded the respiratory and cardiovascular cases from the data, the associations remained robust, with PRs of 1.26 (95% CI: 1.17–1.36) per IQR increase in PM_1_.

## 4. Discussion

The current study is the first to show that long-term exposures to PM and NO_2_ are significantly and consistently associated with decreased haemoglobin levels and increased prevalence of anaemia in a large scale survey of older adults in China. In addition, we observe that among current smokers, the adverse effects of PM_10_ and PM_1_, but not NO_2_, on haemoglobin levels were even stronger (p < 0.01). Moreover, alcohol consumption was shown to enhance this link for PM_1_ and NO_2_ (p < 0.01), but not for PM_2.5_ or PM_10_. Dose-response curves for all associations demonstrated significant non-linearity. We further estimated that 4.4% of anaemia cases in the study population could be attributed to NO_2_ exposure and that 26% of the anaemia cases observed in the study could have been averted if NO_2_ in the study locations met the WHO guidelines.

Our findings of the air pollution impact on haemoglobin levels in older adults are consistent with the results of a recent US study. Honda et al. found that an IQR (3.9 μg/m^3^) increase in one-year moving average PM_2.5_ was linked to significantly higher anaemia prevalence (PR 1.33; 95% CI: 1.23–1.45) and decrement in haemoglobin concentrations of 0.81 g/dL (95% CI: 0.75–0.87). Similarly, an IQR (9.6 ppb) rise in NO_2_ was associated with an increased prevalence of anaemia (PR 1.43; 95% CI: 1.25–1.63) and a decrease in average haemoglobin of 0.81 g/dL [43]. Our findings are also consistent with other studies evaluating the associations between long-term air pollutant concentrations with haemoglobin levels in a younger population. A 2006 study of 327 Serbian individuals found that a higher five-year average pollution (sulphur dioxide (SO_2_), soot and lead (Pb) in sediment matters in the air) was linked to an increased risk of anaemia during pregnancy [66]. Meanwhile, Nikolic’ et al. found that children who are exposed to a high level of long-term NO_2_, black smoke and SO_2_ had lower haemoglobin levels and RBC counts compared to those who lived in less polluted areas [67]. 

Studies investigating associations between short-term air pollution exposures and haemoglobin levels show inconsistent results. For instance, in a UK study of acute air pollution exposures in 112 older adults, three-day average PM_10_ exposure was associated with an 0.44 g/dL decrease in average haemoglobin per 100 μg/m^3^ increase in PM_10_ [58]. Similar findings were reported in a cross-sectional study in India, in which 24-hour exposures of PM_10_, NO_2_ and SO_2_ were significantly associated with lower haemoglobin concentrations (p < 0.05) [68]. In contrast, a study by Sørensen et al. of 50 young adults (aged 20–33 years) found that PM_2.5_ and black carbon exposures were associated with higher haemoglobin and erythrocyte counts, with a two-day average air pollution exposure associated with a 2.3% increase in RBC number, and a 2.6% increase in haemoglobin per 10 μg/m^3^ increase in PM_2.5_ in women, but not in men. In a study of children aged 6–59 months living in one of the most air-polluted cities of Latin America (Lima, Peru), investigators found that high levels of outdoor PM_2.5_ (25–50 μg/m^3^) were associated with a 39.6% of anaemia prevalence (95% CI: 39.3–39.9), were conversely males were one of the most affected groups [69].

The inconsistency of the findings from the studies mentioned above on the short-term air pollution exposures may reflect several factors. First, all these studies have a small geographically distinct and thus non-representative sample size, which limits the generalisability of the findings to other populations. Second, short-term pollutant exposure likely affects haematologic and haematopoietic physiology differently than long-term exposures [70], and older adults are possibly affected differently than younger individuals. 

Air pollution is known to cause a chronic systemic inflammatory response [71]. This, in turn, may lead to decreased erythropoietin secretion by the kidneys as well as increased endogenous erythropoietin resistance in the bone marrow [72,73]. Together, these two factors can lower the production of RBCs and result in decreased haemoglobin levels [72]. Another possible explanation is that air pollution can induce IL-6 gene expression in alveolar macrophages, which has been previously shown to upregulate hepatic hepcidin production [74]. Increased hepcidin, an iron regulatory protein, can decrease gastrointestinal iron absorption, contributing to iron deficiency anaemia [75]. These mechanisms related to chronic inflammation and perturbed iron regulation, taken together, might explain how long-term, although not short-term, air pollution exposures may contribute to decreased haemoglobin levels and anaemia [43]. 

We found that for PM_10_ and PM_2.5_, the adverse effects of air pollution on haemoglobin levels were amplified in smokers (−0.78 ± 0.10 g/dL) as compared to non-smokers (−0.61 ± 0.08 g/dL). This finding is inconsistent with findings by Honda et al. (2018), who found that NO_2_ exposure, but not PM, was associated with higher magnitude associations in older US adults. It is possible that local immune response differences between smokers and non-smokers may underlie our result, which makes smokers more susceptible to air pollution exposures. Additionally, effect modification was observed for alcohol consumption, where increases in PM_1_ and NO_2_ among those who reported ever consuming alcohol were associated with higher magnitude associations relative to those who reported never consuming alcohol. This outcome is also unexpected, as prior literature has found that alcohol consumption is often associated with increased haemoglobin concentrations [76]. Also, we found greater association in the northern regions of china compared to south of china, this outcome is expected because of the spatial distribution of ambient air pollution in China. Previous studies reported the average concentration of PM_2.5_ in north China reached as high as 80–100 μg/m^3^, while in south, the concentration has reduced to 40–70 μg/m^3^ [77,78,79].

Our findings that PM and NO_2_ were associated with decreased haemoglobin levels in Chinese older adults have significant clinical and public health implications. For instance, a 2005 study of older adults (mean 60.2 years) post-myocardial infarction found that just 1 g/dL decrease in haemoglobin levels below 14 mg/dL was linked to 21% increased odds of cardiovascular mortality (95% CI: 12–30%) [80]. This study evaluated the anaemia burden attributable to ambient NO_2_ in a population of Chinese elderly. It also estimated the reduction of the anaemia burden when the NO_2_ levels meet the WHO air quality guidelines. Most of the previously published works usually quantified the association between pollutants and adverse health outcomes. The attributable risk may provide even more information from the point of public health significance. A similar approach has been applied to other research works. In a recent study of six low-to-middle income countries, Guo et al. estimated that 7.11% of stroke cases could be attributable to ambient PM_2.5_ [81]. Moreover, the Global Burden of Disease study reported that 4.2 million (7.6%) deaths globally could be prevented if the standards of the ambient PM_2.5_ were attained [82]. The latter suggests that the impact of air pollution on anaemia may be one way in which air pollution leads to cardiovascular mortality [83]. 

Our study has a number of important limitations. First, as a cross-sectional study, we are not able to establish a temporal relationship between the exposures and outcomes of interest. Second, while we used a satellite-based comprehensive exposure model and assigned exposures according to the participants’ addresses at the township level, the possibility of exposure misclassification is high, as we were unable to have more detailed information to inform our exposure assessment, such as the respondents’ activity patterns regarding time spent in traffic and indoors. Third, specific components of PM that may be driving our results, such as transition and/or heavy metals, could not be considered separately. They might have had distinct chemical structures and might be associated with different health concerns. Last, some critical risk factors of anaemia, such as genetic background and cultural dietary differences among various geographic areas, were not taken into account. However, in our multilevel models, we attempted to address this by controlling for the community in which an individual lives. 

Several significant strengths counterbalance these limitations. First, our study encompasses a large, representative population to examine the association between ambient air pollution and anaemia among older adults in China. We collected extensive individual-level information on other risk factors, allowing us to control for individual-level confounders. Second, the use of satellite-based estimates of PM_1_ and PM_2.5_ exposures enabled us to have a virtually complete spatial coverage among the study participants in the absence of air monitoring data in China for PM_2.5_ before the year of 2012. Third, our study is the first to investigate the PM_1_ association with anaemia in any settings globally. The existing air pollution standards in both the United States and China do not include regulations for PM_1_. Therefore, there is an urgent need to fill these gaps in the literature and our study can be used to inform public policies and guidelines in order to protect older adults from hazardous risks associated with PM_1_ air pollution.

## 5. Conclusions

Our study is the first to demonstrate that exposure to ambient air pollution is associated with an increased prevalence of anaemia and reduced levels of haemoglobin in Chinese older adults. Further studies to examine the mechanisms accounting for increased vulnerability to the PM and NO_2_ are warranted. Public policies and guidelines should be improved to protect the ageing population from risks associated with air pollution. 

## Figures and Tables

**Table 1 ijerph-17-03209-t001:** Characteristics of the older adults’ study participants in the 64 communities of Study of Global Ageing and Adult Health (SAGE) China Wave 1.

Variables	Non-Anaemic	Anaemic	Total	P Value for Difference
PM_10_ (ug/m^3^, mean (SD))	88.80 (30.11)	86.71 (21.18)	88.21 (27.91)	<0.001
PM_2.5_ (ug/m^3^, mean (SD))	52.64 (17.60)	52.52 (13.26)	52.61 (16.49)	0.74
PM_1_ (ug/m^3^, mean (SD))	42.22 (13.10)	44.45 (13.70)	42.85 (13.31)	<0.001
NO_2_ (ppb, mean (SD))	28.36 (11.45)	32.26 (14.48)	29.46 (12.50)	<0.001
Age (years, mean (SD))	62.62 (9.09)	64.03 (9.74)	63.02 (9.30)	<0.001
BMI (kg/m^2^, mean (SD))	24.28 (5.41)	23.42 (4.30)	24.04 (5.14)	<0.001
Sex				0.34
Male (n, %)	3667 (48.09)	1405 (47.05)	5072 (47.8)	
Female (n, %)	3958 (51.91)	1581 (52.95)	5539 (52.2)	
Smoker				0.17
Ever (n, %)	2623 (34.61)	987 (33.21)	3610 (34.21)	
Never (n, %)	4956 (65.39)	1985 (66.79)	6941 (65.79)	
Alcohol				0.80
Ever (n, %)	2341 (30.94)	911 (30.68)	3252 (30.87)	
Never (n, %)	5225 (69.06)	2,058 (69.32)	7283 (69.13)	
Education				<0.001
No formal education (n, %)	1802 (23.63)	847 (28.37)	2649 (24.96)	
Primary school (n, %)	3032 (39.76)	1135 (38.01)	4167 (39.27)	
Middle school (n, %)	1470 (19.28)	570 (19.09)	2040 (19.23)	
High school or higher (n, %)	1321(17.32)	434 (14.53)	1755 (16.54)	
Place of residence				<0.001
Rural (n, %)	4514 (59.2)	1225 (41.02)	5739 (54.09)	
Urban (n, %)	3111 (40.8)	1761 (58.98)	4872 (45.91)	
Physical activity				<0.001
Low level (n, %)	2485 (32.78)	828 (27.86)	3313 (31.4)	
Moderate level (n, %)	2073 (27.35)	863 (29.04)	2936 (27.82)	
High level (n, %)	3022 (39.87)	1281 (43.1)	4303 (40.78)	
Nutrition				<0.001
Insufficient intake (n, %)	3278 (42.99)	1455 (48.73)	4733 (44.6)	
Sufficient intake (n, %)	4347 (57.01)	1531 (51.27)	5878 (55.4)	
Type of fuel used at home				<0.001
Clean (n, %)	3795 (50.16)	1785 (59.84)	5580 (52.9)	
Unclean (n, %)	3771 (49.84)	1198 (40.16)	4969 (47.1)	
History of Diabetes				0.50
Yes (n, %)	467 (6.18)	194 (6.54)	661 (6.29)	
No (n, %)	7084 (93.82)	2772 (93.46)	9856 (93.71)	
History of Hypertension				0.09
Yes (n, %)	2069 (27.66)	770 (26.01)	2839 (27.19)	
No (n, %)	5412 (72.34)	2190 (73.99)	7602 (72.81)	
History of Chronic lung diseases				0.42
Yes (n, %)	618 (8.17)	257 (8.66)	875 (8)	
No (n, %)	6942 (91.83)	2711 (91.34)	9653 (92)	

PM_10_, particulate matter with an aerodynamic diameter less than or equal to 10 μm; PM_2.5_, particulate matter with an aerodynamic diameter less than or equal to 2.5 μm; PM_1_, particulate matter with an aerodynamic diameter less than or equal to 1 μm and NO_2_ nitrogen dioxide. SD: standard deviation; BMI: body mass index.

**Table 2 ijerph-17-03209-t002:** Distributions of 3-year average concentrations of air pollutants in 64 communities in China.

Study Region	PM_10_ (μg/m^3^)	PM_2.5_ (μg/m^3^)	PM_1_ (μg/m^3^)	NO_2_ (ppb)
Guangdong	80.32 (2.91)	51.63 (2.67)	37.39 (6.41)	24.30 (4.87)
Hubei	108.36 (2.37)	67.64 (2.70)	46.77 (6.71)	31.52 (1.58)
Jilin	74.70 (3.56)	42.19 (1.85)	39.80 (6.32)	20.20 (4.73)
Shaanxi	91.24 (19.35)	48.84 (10.41)	42.44 (12.00)	24.40 (5.90)
Shangdong	135.85 (14.25)	71.28 (11.18)	38.00 (10.86)	32.60 (9.85)
Shanghai	100.55 (1.12)	69.68 (.77)	60.25 (9.10)	46.31 (1.60)
Yunnan	47.10 (5.14)	27.90 (4.99)	37.70 (13.18)	19.19 (3.49)
Zhejiang	83.86 (11.40)	50.96 (7.85)	35.10 (12.60)	26.96 (11.94)
Mean (SD)	91.11 (28.95)	54.02 (17.02)	42.90 (13.13)	28.97 (11.31)
Median (IQR)	93.79 (31.15)	55.62 (26.14)	41.54 (24.12)	24.18 (22.42)

PM_10_: particulate matter with an aerodynamic diameter less than or equal to 10 μm; PM_2.5_: particulate matter with an aerodynamic diameter less than or equal to 2.5 μm; PM_1_: particulate matter with an aerodynamic diameter less than or equal to 1 μm; NO_2_: nitrogen dioxide. IQR was computed by subtracting the 25th percentile from the 75th percentile. SD: standard deviation; IQR: interquartile range.

**Table 3 ijerph-17-03209-t003:** Association between prevalence ratio of anaemia and 3-year moving average of pollutant.

	PR (95% CI)			
	PM_10_	PM_2.5_	PM_1_	NO_2_
**Unadjusted**	0.94 (0.91–0.97)	0.99 (0.95–1.03)	1.24 (1.17–1.31)	1.46 (1.39–1.54)
**Adjusted ^†^**	1.05 (1.02–1.09)	1.11 (1.06–1.16)	1.13 (1.06–1.20)	1.42 (1.34–1.49)

3-Years average IQR: PM_10_, 31.15 μg/m^3^; PM_2.5_, 26.14 μg/m^3^; PM_1_, 24.12 μg/m^3^; NO_2_, 22.42 μg/m^3^. ^†^ Adjusted model includes pollutant, age, sex, tobacco use, physical activity, education, body mass index (BMI), alcohol, place of residence, household income, diabetes history, hypertension history, chronic lung disease history, indoor fuel, fruit and vegetables consumption. PM_10_: particulate matter with an aerodynamic diameter less than or equal to 10 μm; PM_2.5_: particulate matter with an aerodynamic diameter less than or equal to 2.5 μm; PM_1_: particulate matter with an aerodynamic diameter less than or equal to 1 μm; NO_2_: nitrogen dioxide; CI: confidence interval; IQR: interquartile range.

**Table 4 ijerph-17-03209-t004:** Association between haemoglobin levels g/dL and 3-year moving averages of pollutant.

	β (95% CI)			
	PM_10_	PM_2.5_	PM_1_	NO_2_
**Unadjusted**	−0.77 (−0.89, −0.65)	−1.06 (−1.22, −0.91)	−0.98 (−1.08, −0.94)	−1.98 (−2.11, −0.89)
**Adjusted ^†^**	−0.53(−0.67, −0.38)	−0.52(−0.71, −0.33)	−0.55 (−0.69, −0.41)	−1.71 (−1.85, −1.57)

3-Years average IQR: PM_10_, 31.15 μg/m^3^; PM_2.5_, 26.14 μg/m^3^; PM_1_, 24.12 μg/m^3^; NO_2_, 22.42 μg/m^3^. ^†^ Adjusted model includes pollutant, age, sex, tobacco use, physical activity, education, body mass index (BMI), alcohol, place of residence, household income, diabetes history, hypertension history, chronic lung disease history, indoor fuel, fruit and vegetables consumption. PM_10_: particulate matter with an aerodynamic diameter less than or equal to 10 μm; PM_2.5_: particulate matter with an aerodynamic diameter less than or equal to 2.5 μm; PM_1_: particulate matter with an aerodynamic diameter less than or equal to 1 μm; NO_2_: nitrogen dioxide; CI: confidence interval.

**Table 5 ijerph-17-03209-t005:** Population attributable risk (PAR) and population attributable fraction (PAF) of nitrogen dioxide 3-year annual moving average.

	N0_2_ 40 µg/m^3^ or Less Annual Average
^†^ Adjusted PAR (95% CI)	4.4% (3.6–5.2)
^†^ Adjusted PAF (95% CI)	26.14% (14.94–35.86)

^†^ Adjusted for age, sex, tobacco use, physical activity, education, body mass index (BMI), alcohol, place of residence, household income, diabetes history, hypertension history, chronic lung disease history, indoor fuel, fruit and vegetables consumption. CI: confidence interval.

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
