# Peer review of "Ambient Air Pollution Exposure Association with Anaemia Prevalence and Haemoglobin Levels in Chinese Older Adults"

_ijerph, 2020, doi:10.3390/ijerph17093209_

Round 1
Reviewer 1 Report
I want to thank the authors for their hard work in their rewrite.
I have only 1 small item you might think about changing. Thinking it has been 18 years since the WHO data point, is there a more current number that may make the information more impactful.
Reviewer 2 Report
The research method and analysis method of this research have credibility. The amount of data is sufficient and the source is reliable and representative. The issues considered are more comprehensive. And this version is better than the last one. It has been modified according to the comments. It is suggested that this manuscript be accepted.
This manuscript is a resubmission of an earlier submission. The following is a list of the peer review reports and author responses from that submission.
Round 1
Reviewer 1 Report
This is a well done epidemiological study based upon a multinational effort that shows quite clearly that exposure of an adequately sized population of older Chinese individuals to ambient air pollution is associated with an increased prevalence of anemia in the frame of a WHO sponsored study. The authors found that all the measured pollutants (PM and NO2) were associated in terms of degree of exposure to the presence of anemia. This study is well done and the results biologically plausible, with no specific criticisms by me on the resulting findings. Even though it is true that no similar study was previously done on the at-risk older Chinese population, this reviewer finds that this manuscript contains very little new knowledge with global implications. Thus low priority, unless you have much space available in the journal.
Reviewer 2 Report
I think the authors did a great job comparing air pollution exposure and anemia. I think that it would be neat with the data present in WHO to look at all age groups in all the areas. Being so narrow in the cross sectional study it makes it difficult (as you pointed out) to generalize to a larger group making external validity difficult. I think when you think about internal validity knowing that the older population are more anemic having the total age groups might be interesting in the future. Also since the data is longitudinal maybe you can find the data from those that were non-anemic and follow them through time to see when and if they become anemic.
Overall good job.
Reviewer 3 Report
The research method and analysis method of this research have credibility. The amount of data is sufficient and the source is reliable and representative. The issues considered are more comprehensive. The analysis of the study is thorough and innovative. there are still a few suggestions:
- Each table of supplementary materials can be the same as the table of the text, mentioned in the article, and arranged in order. For example, table S4 appears in line 282, but S2 appears in line 304. The order of table numbers should be adjusted accordingly. And for example S3 and so on are not mentioned in the article. Adding these tables to the explanation of the text can make the reader realize the function of the results of these tables in the text and play a better role in explaining.
- The pollution situation and industrial structure of different provinces in China are very different. In the process of preliminary analysis, model analysis and interpretation of final results, they can be classified according to provinces. Try to see if the degree of pollution in each province is related to the incidence of the population from a spatial perspective, so as to further support the conclusion and explore the reason
- The data section mentions the simulation methods of various pollutants. The original data, such as years, monitoring data (the number of monitoring stations are also different), are different. Will differences in the simulation results affect the differences in the final regression analysis results?
- Sensitivity analysis only analyzes the years before the baseline survey, but you can consider the years after the study was conducted(2007-2010), and is not limited to the impact of pollution exposure before survey.